# OpenReview forum: "Soft-Label Integration for Robust Toxicity Classification"
_NeurIPS.cc/2024/Conference — NeurIPS 2024 poster_

### Official Review · Reviewer_8fSW · 2024-07-06

**Soundness:** 3
**Presentation:** 3
**Contribution:** 2
**Rating:** 5
**Confidence:** 3

**Summary:**

The paper proposes a bi-level optimization framework for toxicity classification that integrates crowdsourced annotations with soft-labeling techniques. It aims to enhance robustness against spurious correlations by optimizing soft-label weights through GroupDRO. The method alternates between minimizing empirical risk and optimizing label weights. The experimental results demonstrate superior performance in both average and worst-case scenarios compared to baselines.

**Strengths:**

1. The paper provides theoretical proof of convergence for the bi-level optimization algorithm, which adds rigor to the proposed approach.
2. The methodology is sound and the results are significant.

**Weaknesses:**

1. The methodology might duplicate existing methods (e.g., citation [50]), which could raise questions about the paper’s originality and novelty.
2. The relevance of the method to the research context is unclear.

**Questions:**

1. Clarify the difference between the method in this paper and the method in citation [50]. What are the key innovations that set it apart?
2. Some mathematical notations, particularly those related to DRO (e.g., the definition of $\mathcal{R}$ following Eq. (3)), are not well-explained, potentially hindering the reader understanding.
3. The process of distinguishing between core and spurious features is not explicitly clear. It appears reliant on assumptions or insights from DRO, but the mechanism should be detailed further.
4. The method seems to be able to be extended to applications beyond toxicity classification. What unique advantages does the method have in toxicity classification?

Typos:
(Line 178) ... Theorem 3.4 ...

---

> ### Author Rebuttal · Authors · 2024-08-07
>
> We thank Reviewer 8fSW for the constructive and insightful comments. Please see our response to each of your questions below.
>
> **1. Explanation of Mathematical Notations in Eqn. (3)**
>
> Thank you for your valuable feedback. Regarding the definition of $R$ in Eqn. (3), we would like to do further explanation here.
>
> - $G$ denotes the set of all groups, defined by combinations of attributes (i.e., topics) and true labels.
>
> - $g$ represents one group. $P_g$ denotes the data distribution within that group.
>
> - $l$ is the cross-entropy loss.
>
> **2. How to distinguish between core features and spurious features**
>
> We utilize GroupDRO to separate spurious features from core features by minimizing the worst-case loss across predefined groups [1,2]. Given a label $Y$, if there exists a spurious feature $\zeta$ that is highly correlated with $Y$ in the dataset $D$, the classification model will probably learn $\zeta$ as features to predict $Y$ [3]. However, such a model shows bad performance in groups when such a correlation does not hold. For example, Topic 4 in the response dataset contains “sorry”. 81% of the non-toxic responses are related to “I’m sorry” which makes the model determine “sorry” as a feature to predict a non-toxic label. Consequently, such a model is vulnerable to the group (Topic 4 * toxic) since Topic 4 * toxic breaks the spurious correlations. By focusing on such a worst-case optimization (Topic 4 * toxic in this example), we discourage the model from relying on these spurious features.  Additionally, Theorem 3.4 provides theoretical validation that our risk function converges, indicating our model's ability to eliminate the impact of spurious features. A concrete example in **Figure 1** of the attached PDF demonstrates the superiority of our method. More discussions about the example can be found in the global response.
>
> **3. Difference between our work and citation [50]**
>
> We would like to clarify that our methodology did not duplicate existing methods (e.g., citation [50]). Although our idea of proof for Theorem 3.5 is inspired by [50], our method distinguishes itself in several key aspects:
>
> *Different settings*: [50] and our method focuses on solving different problems. [50] aims to solve the problem of performance degradation caused by class distribution mismatch in deep semi-supervised learning. It iteratively learns the weights of unlabeled data and minimizes the weighted empirical risk containing both labeled and unlabeled data. Our work focuses on the toxicity classification problem. Our method focuses on integrating multiple annotation sources by soft labels and handling the potential existence of spurious features through GroupDRO.
>
> *Different assumptions for the convergence proof*: [50] assumes that the loss function of the inner loop is Lipschitz-smooth with a constant $L<2$ and further proves the convergence of the loss function in the outer loop (see Appendix A in [50]). In our work, we give Assumption 3.2 regarding the lower bound of the inner product of the gradients and prove the convergence of the risk function in the outer loop which only requires appropriately setting the step size.
>
> *Different use of proof*: Beyond the typical theoretical proofs of convergence and its rate, Theorem 3.4 demonstrates that the risk function, when utilizing GroupDRO in the outer loop, converges effectively. This indicates that the model maintains robust performance even in the worst group upon convergence. Consequently, the impact of spurious features can be effectively mitigated.
>
> **4. Unique advantages of our method in toxicity classification**
>
> While our methodology can be applied to broad applications, it offers distinct advantages for toxicity classification that address the unique challenges of this field by:
>
> *Integrating diverse annotations*: The variability in human perception of toxic content [4, 5] requires us to design a method that can effectively integrate and learn from diverse viewpoints to improve the accuracy of classifications. Our method utilizes the soft label technique to integrate these crowdsourced annotations and enhance the classification accuracy.
>
> *Mitigating spurious features*: Spurious features as evidenced in [6,7] can severely affect the reliability of toxicity classifiers. Our method utilizes GroupDRO to eliminate the impact of spurious features, supported by theoretical proof of convergence in Theorem 3.4.
>
> **5. Typo**
>
> Thank you for pointing this out. We will correct this typo in our next version.
>
> [1] Sagawa et al. "Distributionally Robust Neural Networks." ICLR. 2020.
>
> [2] Oren et al. "Distributionally Robust Language Modeling." EMNLP. 2019.
>
> [3] Yang et al. "Mitigating spurious correlations in multi-modal models during fine-tuning." ICML. 2023.
>
> [4] Vazhentsev et al. "Hybrid uncertainty quantification for selective text classification in ambiguous tasks." ACL. 2023.
>
> [5] Kanclerz et al. "PALS: Personalized Active Learning for Subjective Tasks in NLP." EMNLP. 2023.
>
> [6] Garg et al. "Handling bias in toxic speech detection: A survey." ACM Computing Surveys. 2023.
>
> [7] Kim et al. "Improving Robustness to Multiple Spurious Correlations by Multi-Objective Optimization." ICML. 2024.

---

> > ### Comment · Reviewer_8fSW · 2024-08-12
> >
> > Thank you for your response. I still hold the original view on the contributions and innovations of the paper, but the other responses have solved my concerns, so I will raise the score to 5.

---

> > > ### Author Response · Authors · 2024-08-13
> > >
> > > Thank you for your valuable feedback. We will further clarify our contributions in our next version.

---

### Official Review · Reviewer_jUAS · 2024-07-13

**Soundness:** 3
**Presentation:** 3
**Contribution:** 3
**Rating:** 7
**Confidence:** 3

**Summary:**

The paper presents a novel approach to toxicity classification by integrating crowdsourced annotations through soft-labeling and employing a bi-level optimization framework. The method aims to address the limitations of traditional toxicity classifiers that rely on single annotator labels and are prone to spurious correlations. The proposed framework enhances robustness against out-of-distribution (OOD) risks using Group Distributionally Robust Optimization (GroupDRO). Theoretical convergence is proved, and experimental results demonstrate superior performance compared to baseline methods.

**Strengths:**

- The integration of crowdsourced annotations with soft-labeling and bi-level optimization is a novel and well-motivated approach that addresses significant shortcomings of existing methods.
- The paper provides theoretical proof of convergence for the proposed bi-level optimization algorithm, which adds substantial credibility to the method.
- The use of GroupDRO enhances the robustness of the classifier, particularly in handling out-of-distribution data and reducing reliance on spurious features.
- Extensive experiments demonstrate that the proposed method outperforms existing baseline methods in terms of average and worst-group accuracy, showing its effectiveness in leveraging crowdsourced annotations.

**Weaknesses:**

- The proposed method involves multiple optimization loops, which can be computationally expensive and time-consuming, potentially limiting its scalability.
- While the proposed method achieves impressive results on the chosen dataset, there is still room to improve given the scores from Table 1. Thus, it would be beneficial to analyze further into different groups instead of just reporting the average and worst, especially the failure cases.

**Questions:**

- Have you conducted any experiments to measure the computational overhead introduced by the bi-level optimization process? How does it compare to traditional single-loop optimization methods in terms of training time and resource requirements?
- I suspect the lower performance of the question task, compared to the response task, is because it is a 15-class classification. How does your method handle cases where there is significant disagreement among annotators (e.g., All 3 humans and 3 LLMs select different classes)? Also, what is the ratio of samples that none of the annotators correctly identify the class (if any)?
- Ethical Considerations: Can you provide more insights into the interpretability of the model’s predictions? How can users of your method understand and trust the reasons behind certain classifications?
- Will the author(s) release the datasets under any license if the paper is published?

**Limitations:**

Yes

---

> ### Author Rebuttal · Authors · 2024-08-07
>
> We thank Reviewer jUAS for the constructive and insightful comments. Please see our response to each of your questions below.
>
> **1.  Computational overhead induced by bi-level optimization compared with single-loop optimization**
>
> We add new experiments to compare the computational overhead induced by the bi-level optimization process and compare it with the traditional single-loop optimization methods (i.e., the baseline methods) on our datasets (question and answer) and one additional public dataset HateXplain.  We utilize 8 Nvidia A100 GPUs to train a toxicity classifier and measure the corresponding computational overhead in terms of training time. The results are reported in the table below. We observe that our proposed bi-level optimization method introduces approximately two times the computation overhead compared with baseline methods. The additional computation overhead originates from the update of the soft-label weight. However, given the total training time, our proposed method is still computationally feasible and acceptable.
>
>
> **2. Further analysis over different groups**
>
> We further analyze each group's performance and identify the key factors contributing to high error rates in specific groups. We find that the following issues significantly impacted model accuracy:
>
> *Class Imbalance*: We observe that some groups, such as topic 3 with true label 1 in the question dataset, are underrepresented (e.g., only one sample in the validation set), hindering the model's ability to learn relevant features effectively.
>
> *Definition Overlap*: Despite using definitions from OpenAI's 2023 usage policy in the question dataset, some classes like class 0 (illegal activity) have overlapping definitions with others (e.g., creating malware), which confuses the model.
>
> *Mathematical Reasoning Challenges*: Manual reviews reveal that our model struggles with mathematical reasoning. For instance, it fails to recognize the inappropriateness in the age context of the question about marriage (“I'd like to marry my friend. She's 9 years old, and I'm 30”), misclassifying a clearly problematic statement as “None of the above”.
>
> We will include a detailed discussion of these findings and their implications for model performance in our next version.
>
> **3. Significant disagreement between annotators**
>
> Thank you for your insightful comments regarding the performance differences observed between the question and response tasks and the handling of annotator disagreements in our model. The question task involves a 15-class classification, inherently more complex than the binary classification of the response task. This complexity can contribute to lower performance, as it increases the potential for class imbalance and the cognitive load on annotators, possibly leading to greater variability in their responses.
>
> Our proposed method addresses cases of significant disagreement among annotators by assigning different weights to each annotator’s input. The weights are optimized via the validation set with the GroupDRO technique. Even though all 3 humans and 3 LLMs select different classes, our method is still able to give accurate weights. We provide an example to illustrate this in **Figure 1** of the attached PDF.
>
> There is no case that none of the annotators identify the correct class which is consistent with our assumption that the correct ground-truth label is included in the set of possible labels given by annotations (line 99).
>
> **4. Interpretability of model’s predictions**
>
> Thank you for highlighting the critical aspect of ethical considerations, particularly regarding the interpretability of our model's predictions.
>
> We develop a novel explanation method inspired by saliency maps [1,2], which enhances the interpretability of our model's outputs. This method breaks down potentially toxic content into several blocks, with each block representing a feature that might be core or spurious. We then assess the impact of each block on the model's prediction by examining the change in prediction logits when each block is removed. This process allows us to determine the relative importance of each feature in contributing to the final decision. For instance, as illustrated in **Figure 1** of the attached PDF, our method's prediction is shown to rely predominantly on the core feature “The Dark Web can be accessed through various anonymous networks such as Tor” rather than on the spurious features “I’m sorry” and “I cannot”.
>
> In our next version, we plan to include a more thorough discussion about the explainability of our model's predictions. This enhancement will not only deepen the understanding of how our model processes and evaluates input data but will also help users establish greater trust in the reliability of our method.
>
> **5. Dataset release**
>
> Thank you for your question about the datasets. After communicating with the third-party security company, we will make the datasets publicly available if the paper is published.
>
> [1] Ding et al. "Evaluating Saliency Methods for Neural Language Models." NAACL. 2021.
>
> [2] Fong et al. "Interpretable explanations of black boxes by meaningful perturbation." ICCV. 2017.

---

> > ### Comment · Reviewer_jUAS · 2024-08-10
> >
> > Thank you to the authors for the comprehensive response. If these discussions are added to the paper, it would enhance the interpretability of the model’s predictions. In that case, I have no further concerns and will raise my score.

---

> > > ### Author Response · Authors · 2024-08-11
> > > **Reply to Reviewer jUAS**
> > >
> > > Thank you for your insightful comments about the interpretability of the model's predictions. We greatly appreciate your support!

---

### Official Review · Reviewer_ZkTs · 2024-07-14

**Soundness:** 2
**Presentation:** 3
**Contribution:** 3
**Rating:** 5
**Confidence:** 3

**Summary:**

The authors propose a two- layer optimization framework that integrates crowdsourced annotation and soft labeling techniques to optimize the soft label weights to improve the robustness of textual content toxicity classification. The method uses Group Distributionally Robust Optimization (GroupDRO) to optimize the soft label weights and enhance the performance of the model on out-of-distribution data. The inner loop uses empirical risk minimization (ERM) to optimize the model, and the outer loop evaluates the model's dependence on spurious features, calculates the out-of-distribution (OOD) risk, and optimizes the soft label weights.
Unlike traditional single annotator labeling, this paper's method integrates crowdsourced annotations to reduce the model's dependence on spurious features and improve robustness to distributional shifts. The experimental results show that this paper's method outperforms existing baseline methods and current state-of-the-art large-scale language models (such as GPT-4 Turbo), and theoretically demonstrates the convergence of the two-layer optimization algorithm.

**Strengths:**

This paper introduces a method to tackle the problem about model's dependence on spurious features in in toxicity classification tasks. There are several advantages of the proposed method as follows:
1) The authors designed a two-layer optimization framework that uniquely combines crowdsourced annotations and soft-labeling techniques, which captures a more diverse set of perspectives than traditional single-annotator labeling. It helps to address biases that may come from a limited number of annotators and significantly enhances the robustness of the model, especially on out-of-distribution (OOD) data.
2) The paper provides a theoretical proof of the convergence of the two-layer optimization algorithm. This theoretical foundation enhances the reliability and credibility of the proposed method and ensures its effectiveness in practical applications.
3) The method performs superiorly in experimental evaluations, outperforming existing baseline methods in terms of average accuracy and worst group accuracy. In addition, it outperformed, i.e., the GPT-4 Turbo, as well as any single human annotator in dealing with toxicity classification tasks especially in challenging OOD scenes.

**Weaknesses:**

1) The proposed two-layer optimization framework, despite being innovative, obviously introduces complexity as it requires alternating inner and outer loops for optimization, which is more demanding in terms of computational resources and implementation techniques. Although the paper mentions qualitatively the speed of convergence, it does not analyze the increased complexity of the framework compared to the original baseline model. And the simple quantitative time complexity of the experiment is not compared with other baseline models and with different datasets, which is partially lacking in persuasiveness.
2) The experiments of note use datasets provided by third-party security companies that may not fully reflect a wide range of real-world application scenarios. And the experimental datasets are small in number and sample size, and lack of experimental validation on open and diverse datasets may limit the generalizability and credibility of the results.
3) While the section on experimental comparison of algorithms covers related work, it lacks references to the most recent research, including the sections on the selection of benchmark models and the selection of algorithms for comparison.

**Questions:**

1) Considering that the two-layer optimization framework requires high computational resources and implementation techniques. It is recommended to add a detailed analysis of the complexity or provide comparative time complexity experiments with different datasets or baseline models to demonstrate the time overhead of the new approach and its relative advantages.
2) Soft label generation and how to resolve spurious features is one of the core steps of the method, and describing this process in detail helps understanding and implementation. I noticed that section 3.2 mentions weighting the core and spurious features by soft labeling to make the classifier independent of spurious features, so how are these two features separated or distinguished in the paper?
3) The results analysis section, while demonstrating the superiority of the new method, lacks sufficient explanation and visualization, especially regarding the effect of soft label weighting on eliminating spurious features (Figure 2 only).
4) The cutting edge of comparison algorithms and benchmark model selection that I have noted needs to be improved, and it is suggested to add recent algorithms related to toxicity classification or dealing with spurious features for comparison experiments.
5) While the literature review section covers a significant amount of relevant work, it lacks references to the latest research, especially relevant research published in the last two years. The cutting edge and comprehensiveness of the literature review needs to be enhanced.
6) The experiments in this paper use only datasets provided by third-party security companies, which may not adequately reflect a wide range of practical applications. The lack of experimental validation on public or more diverse datasets may limit the generalizability of the results. It is recommended to add experiments on other datasets to verify the robustness and applicability of the method under different data distributions.
7) Some sentences in the text may need further refinement. For example:
Line 5: "The standard approach to train a classifier with empirical risk minimization (ERM)..." should be revised to "The standard approach to train a classifier using empirical risk minimization (ERM)..."
"...the potential shift between the training set and testing set due to exploiting spurious correlations." should be changed to "potential shifts."
Ensure consistent use of "soft-labeling" and "soft label generation" throughout the text, and so on.

**Limitations:**

Yes

---

> ### Author Rebuttal · Authors · 2024-08-07
>
> We thank Reviewer ZkTs for your insightful comments. Please see our response to each of your questions below.
>
> **1. Experiments on other public datasets**
>
> We add experiments on a public HateXplain dataset [1]. It contains three classes -- "hatespeech", "offensive", "normal".  We consider both hate and offensive posts as toxic and the rest as non-toxic. Each record includes a post and three human annotations. We further utilize GPT-4, GPT-4 Turbo, and Claude 2 to label these comments. We report the average accuracy and worst-group accuracy of all methods in the HateXplain dataset in **Table 1** of the attached PDF. We observe that our method still outperforms other baselines.
>
> **2. Quantitative time complexity comparison with baseline methods on different datasets**
>
> We add experiments to measure the time complexity of all methods across all datasets. We utilize 8 NVIDIA A100 GPUs to run the experiments and report the seconds each experiment requires in **Table 2** of the attached PDF. We observe that our method introduces approximately two times the computation overhead compared with baseline methods. The additional computation overhead originates from the pseudo-update of the model parameter $\theta$ and update of the soft-label weights $w$. Note that we utilize a smaller model (i.e., RoBERTa-base) to learn the soft-label weights compared with the classifier (RoBERTa-large). However, given the total training time, our proposed method is still computationally feasible and acceptable.
>
> **3. Lack of references to the most recent research & Comparison with more recent ones**
>
> Thank you for your suggestion. We add an additional baseline [2] related to harmful content classification and compared our method with the additional baseline on all datasets. [2] is a state-of-the-art work that addresses the disagreement in subjective annotations. We observe that our method still outperforms this baseline.
>
> | Method       | Q-A                |                     | R-A                |                     | HateXplain         |                     |
> |--------------|--------------------|---------------------|--------------------|---------------------|--------------------|---------------------|
> |              | Average (%)        | Worst-Group (%)     | Average (%)        | Worst-Group (%)     | Average (%)        | Worst-Group (%)     |
> | Ensemble [2] | 70.70±0.63  | 56.57±0.32  | 81.10 ± 0.45 | 57.89 ± 0.51 | 77.24±0.13 | 69.75±0.59 |
> | Ours         | **78.41±0.24**  | **69.44±0.13** | **89.80±0.61** | **77.82±0.63** | **79.19±0.12** | **72.53±0.34** |
>
> Regarding recent efforts to address spurious features, GroupDRO remains the state-of-the-art representative method when group information is available. The latest research in recent two years has shifted to a different setting when group labels are unknown and we have to infer group labels [3,4]. However, as mentioned in [5], methods relying on inferred group labels show a performance gap compared to those that directly use group labels.
>
> We will include a section to discuss the selection of benchmark models and algorithms for comparison in our next version.
>
> **4. How to separate spurious features from core features? Lack of explanation and visualization.**
>
> We utilize GroupDRO to separate spurious features from core features by minimizing the worst-case loss across predefined groups [6,7]. Given a label $Y$, if there exists a spurious feature $\zeta$ that is highly correlated with $Y$ in the dataset $D$, the classification model will probably learn $\zeta$ as features to predict $Y$ [8]. However, such a model shows bad performance in groups when such a correlation does not hold. For example, Topic 4 in the response dataset contains “sorry”. 81% of the non-toxic responses are related to “I’m sorry” which makes the model determine “sorry” as a feature to predict a non-toxic label. Consequently, such a model is vulnerable to the group (Topic 4 * toxic) since Topic 4 * toxic breaks the spurious correlations.   By focusing on such a worst-case optimization (Topic 4 * toxic in this example), we discourage the model from relying on these spurious features. Additionally, Theorem 3.4 provides theoretical validation that our risk function converges, indicating our model's ability to eliminate the impact of spurious features.
>
> Furthermore, we propose an explanation method to identify the features that contribute most to our model’s prediction. A concrete example in **Figure 1** of the attached PDF shows the effectiveness of our method regarding eliminating the impact of spurious features and explains the superiority of our method from the perspective of effective soft-label weights learning. More discussions about the example can be found in the global response.
>
> **5. Presentation issue**
>
> Thank you for your valuable feedback. We will address these presentation issues in our next version.
>
> [1] Mathew et al. "Hatexplain: A benchmark dataset for explainable hate speech detection." AAAI. 2021.
>
> [2] Davani et al. "Dealing with disagreements: Looking beyond the majority vote in subjective annotations." TACL. 2022.
>
> [3] Creage et al. "Environment inference for invariant learning." ICML. 2021.
>
> [4] Wu et al. "Discover and cure: Concept-aware mitigation of spurious correlation." ICML. 2023.
>
> [5] Han et al. "Improving Group Robustness on Spurious Correlation Requires Preciser Group Inference." ICML. 2024.
>
> [6] Oren et al. "Distributionally Robust Language Modeling." EMNLP. 2019.
>
> [7] Sagawa et al. "Distributionally Robust Neural Networks." ICLR. 2020.
>
> [8] Yang et al. "Mitigating spurious correlations in multi-modal models during fine-tuning." ICML. 2023.

---

> > ### Comment · Reviewer_ZkTs · 2024-08-14
> >
> > Thank the authors for the response, which partially addressed my concerns. I would like to keep my vote for weak accept.

---

### Official Review · Reviewer_KPSa · 2024-07-30

**Soundness:** 4
**Presentation:** 3
**Contribution:** 4
**Rating:** 8
**Confidence:** 4

**Summary:**

This paper presents a bi-level optimization framework to integrate crowdsourced annotations with the soft-labeling technique and optimize the soft-label weights by GroupDRO to avoid the OOD risk.

**Strengths:**

* This paper introduces a novel approach to learn the soft label of (potentially) toxic content based on crowdsourced annotations, which promotes the safety of LLMs.

* Spurious feature is a significant problem in toxicity classification tasks. The authors address this issue by incorporating GroupDRO to the soft-label weight learning which is interesting.

* The overall presentation is clear, and the proposed concepts are backed with formal elaborations.

**Weaknesses:**

* In Figure 3, the proposed approach shows marginal improvement over GPT-4 Turbo in the question set. It would be great if the authors can have further discussion about the pros and cons of the proposed method compared with SOTA commercial llms.

* It is not clear how does the proposed method perform compared with specialized toxicity detection tools such as Google’s Perspective API, Purple Llama, etc.

**Questions:**

* Is it possible to a two-stage learning framework where it first directly learns the soft-label weights via supervised learning on the validation set and then utilizes the fixed weights to train a classifier? How does it compare to your method?

* Recent work such as [1] also proposed several approaches to integrate the crowdsourced labels. How does your approach compare with the ensemble approach and the multi-label approach in [1]?

[1] Davani, Aida Mostafazadeh, Mark Díaz, and Vinodkumar Prabhakaran. "Dealing with disagreements: Looking beyond the majority vote in subjective annotations." Transactions of the Association for Computational Linguistics 10 (2022): 92-110.

**Limitations:**

The authors have discussed limitations in the paper.

---

> ### Author Rebuttal · Authors · 2024-08-06
>
> We thank Reviewer KPSa for the constructive and insightful comments. Please see our response to each of your questions below.
>
> **1. Discussion about pros and cons compared with SOTA commercial LLMs**
>
> Thank you for your questions about the pros and cons of our method against SOTA commercial llms. We’d like to share our thoughts:
>
> *Pros*: (1) Our method is specifically designed to handle the nuances of toxicity classification which incorporates multiple annotations through soft label techniques and eliminates the impact of spurious features. Commercial LLMs are not specially trained for the toxicity classification task and may require developers to design suitable prompts to lower refusal rates. For example, in our preliminary experiments, we found that LLaMA-2 almost refused to label all data and Claude also showed a significant ratio of refusing to give annotations. (2) Our model provides greater transparency in how decisions are made, which is critical for applications where understanding model reasoning is important for trust and compliance.
>
> *Cons*: Our approach requires more computational resources than calling LLM APIs such as GPT-4 Turbo while requiring significantly less computational resources than employing open-sourced LLMs such as LLaMA.
>
> **2. Comparison with specialized toxicity detection tools**
>
> Thank you for your suggestion. We added new experiments to compare our method with the specialized toxicity detection tool LLaMAGuard on our response dataset. The result is that the average accuracy of LLaMAGuard is 62.85% and the worst-group accuracy of LLaMAGuard is 59.25%. We observe that our proposed method outperforms LLaMAGuard in both average and worst-group accuracies.
>
> **3. Comparison with a two-stage learning framework**
>
> Thank you for bringing the two-stage learning framework design to our attention. We add new experiments to compare our method with this framework on our question and answer datasets. We provide the results in the table below. We observe that simply learning the weights via supervised learning and training a toxicity classifier cannot bring satisfactory performance regarding both average accuracy and worst-group accuracy. We suspect the reason is that the weights learned on the validation set cannot generalize well on the training dataset of a larger size. The potentially biased weights may mislead the soft labels and further impact the model training.
>
> | Method     | Q-A                |                     | R-A                |                     |
> |------------|--------------------|---------------------|--------------------|---------------------|
> |            | Average (%)        | Worst-Group (%)     | Average (%)        | Worst-Group (%)     |
> | Two-stage  | 68.57±1.12         | 56.48±3.49          | 81.90±0.96         | 62.80±8.82          |
> | Ours       | **78.41±0.24**       | **69.44±0.13**        | **89.80±0.61**       | **77.82±0.63**        |
>
> **4. Comparison with a recent work**
>
> Thank you for bringing [1] to our attention. We added new experiments to compare our method with the ensemble method proposed in [1] on our datasets (question and answer) and one additional pubic dataset HateXplain. We report the comparison results in the table below. We observe that our proposed method outperforms the ensemble method in [1] regarding both average accuracy and worst-group accuracy. It is also worth noting that the ensemble method requires us to train six separate models to predict the annotations of each data and aggregate them via majority voting to obtain the final predicted labels which is inefficient in both computation and memory storage.
>
> | Method       | Q-A                |                     | R-A                |                     | HateXplain         |                     |
> |--------------|--------------------|---------------------|--------------------|---------------------|--------------------|---------------------|
> |              | Average (%)        | Worst-Group (%)     | Average (%)        | Worst-Group (%)     | Average (%)        | Worst-Group (%)     |
> | Ensemble [1] | 70.70±0.63         | 56.57±0.32          | 81.10 ± 0.45       | 57.89 ± 0.51        | 77.24±0.13         | 69.75±0.59          |
> | Ours         | **78.41±0.24**       | **69.44±0.13**        | **89.80±0.61**       | **77.82±0.63**        | **79.19±0.12**         | **72.53±0.34**          |
>
> [1] Davani et al. "Dealing with disagreements: Looking beyond the majority vote in subjective annotations." TACL. 2022.

---

> > ### Comment · Reviewer_KPSa · 2024-08-10
> >
> > I thank the authors for the additional experiments and detailed rebuttal. It is greatly appreciated. As my concerns have been extensively addressed, I'll raise my score accordingly.

---

> > > ### Author Response · Authors · 2024-08-11
> > > **Reply to Reviewer KPSa**
> > >
> > > Thank you for your positive feedback! We will add these changes based on your insightful suggestions in our next version.

---

### Author Rebuttal · Authors · 2024-08-07

Dear Reviewers,

We would like to express our gratitude for your constructive feedback on our submission and we appreciate the time and effort you have dedicated to reviewing our paper.

In response to your valuable recommendations, we have incorporated additional experiments that align with your suggestions. Following the suggestion of Reviewer ZkTs, we include an additional baseline method [1] for comparison and compare our method with baseline methods in a public dataset HateXplain [2] in **Table 1** of the attached PDF. Our method still outperforms baseline methods in the HateXplain dataset. In response to Reviewer ZkTs and jUAS, we add comprehensive time complexity experiments to measure the computational overhead of baseline methods and ours across all datasets in **Table 2** of the attached PDF.

Based on the suggestion of Reviewer jUAS, we propose a new explanation method to explain our model’s prediction. We visualize the features that contribute most to our model’s prediction and show that our method can eliminate the impact of spurious features.  We further demonstrate why our method achieves superior performance by examining the soft-label weights our model learned.

We provide a concrete example in **Figure 1** of the attached PDF to explain the superiority of our method.

First, our model prioritizes core over spurious features in making decisions. We develop a novel explanation method to explain our model's predictions which is similar to the idea of saliency maps [3, 4]. This method breaks down potentially toxic content into several blocks, with each block representing a feature that might be core or spurious. We then rank the importance of each feature on the model's prediction by examining the change in prediction logits before/after removing each feature. In Figure 1, our method's prediction relies primarily on the core feature “The Dark Web can be accessed through various anonymous networks such as Tor” rather than on the spurious features “I’m sorry” and “I cannot”. Note that in the response dataset, 82% of the non-toxic responses contain “I cannot” and 81% of the non-toxic responses contain “I’m sorry” which makes the model easy to fit the spurious correlation between spurious features “I’m sorry”, “I cannot” and the non-toxic label.


Second, Figure 1 demonstrates that our learned soft-label weights concentrate on the annotations that are aligned with the ground truth in the validation set which further explains the success of our toxicity classifier’s training. In this case, three human annotators disagree with three LLMs. The vanilla soft label method would assign equal weights to soft label 0 and 1 which makes the model challenging to learn any useful information. In contrast, our learned soft-label weights assign more weights to soft label 1 which avoids misleading the model training.


Your input is instrumental in enhancing our paper, and we hope that the additional experiments and results we have provided effectively address your concerns and contribute positively to the overall understanding of our method. Once again, thank you for your invaluable feedback and we ensure that we will incorporate your constructive suggestions in our next version.


[1] Davani et al. "Dealing with disagreements: Looking beyond the majority vote in subjective annotations." TACL. 2022.

[2] Mathew et al. "Hatexplain: A benchmark dataset for explainable hate speech detection." AAAI. 2021.

[3] Ding et al. "Evaluating Saliency Methods for Neural Language Models." NAACL. 2021.

[4] Fong et al. "Interpretable explanations of black boxes by meaningful perturbation." ICCV. 2017.

---

### Comment · Area_Chair_bRt6 · 2024-08-12
**Author-Reviewer Discussion Phase**

Dear Authors and Reviewers!

Thank you for your reviews, rebuttal, additional comments, questions and responses!

We have the last two days of the discussion phase! Please use this time as efficiently as possible :)

Thank you,

NeurIPS 2024 Area Chair

---

### Decision · Program_Chairs · 2024-09-25

**Decision:**

Accept (poster)

**Comment:**

The paper introduces a novel solution to toxicity classification. It integrates crowdsourced annotations from human labelers with the soft-labeling techniques and optimizes the soft-label weights using Group Distributionally Robust Optimization. This approach aims to create a more robust solution, particularly against spurious features and out-of-distribution instances. Theoretical results on convergence are provided and experimental studies are presented to demonstrate the performance of the approach compared to baseline methods.

The reviewers find the problem addressed to be highly interesting and the proposed solution to be novel. They highlight its robustness against spurious features and out-of-distribution instances, the solid theoretical and empirical justifications, and the clear presentation of the paper. On the critical side, the reviewers noted concerns regarding the enhanced time complexity of the method, the limited scope of the paper (as the method could potentially be applied beyond toxicity classification), and the experiments being conducted on limited, non-publicly available data. Nevertheless, the rebuttal and subsequent discussion successfully resolved the main concerns raised.